# Leaf and Flower Extracts from the Dwarf Elder (*Sambucus ebulus*): Toxicity and Repellence against Cosmopolitan Mosquito-Borne Diseases Vectors

**DOI:** 10.3390/insects15070482

**Published:** 2024-06-28

**Authors:** Priscilla Farina, Claudia Pisuttu, Camilla Tani, Stefano Bedini, Cristina Nali, Marco Landi, Giulia Lauria, Barbara Conti, Elisa Pellegrini

**Affiliations:** 1Department of Agriculture, Food and Environment, University of Pisa, Via del Borghetto 80, 56124 Pisa, Italy; priscilla.farina@agr.unipi.it (P.F.); claudia.pisuttu@agr.unipi.it (C.P.); camilla.tani@phd.unipi.it (C.T.); stefano.bedini@unipi.it (S.B.); cristina.nali@unipi.it (C.N.); marco.landi@unipi.it (M.L.); giulia.lauria@phd.unipi.it (G.L.); elisa.pellegrini@unipi.it (E.P.); 2Nutrafood, Research Center Nutraceuticals and Food for Health-Nutrafood, University of Pisa, Via del Borghetto 80, 56124 Pisa, Italy; 3Cirsec, Centre for Climatic Change Impact, University of Pisa, Via del Borghetto 80, 56124 Pisa, Italy

**Keywords:** AChE, *Aedes albopictus*, Culex pipiens, larvicidal, OAI, phenylpropanoid

## Abstract

**Simple Summary:**

There has been no scientific evidence of the bioactivity of *Sambucus ebulus* (Adoxaceae) extracts against insects. Therefore, we extracted and chemically characterized the leaves and flowers of *S. ebulus* in methanol and water. The two crude extracts and some of the phenolic compounds and amino acids isolated were tested as larvicides against two cosmopolitan mosquito species, namely the Asian tiger mosquito (*Aedes albopictus*) and the common house mosquito (*Culex pipiens*). To better understand their mode of action, we evaluated the in vitro acetylcholinesterase inhibitor effect of crude extracts on the two mosquito larvae by means of a colorimetric method. In addition, the ovideterrent effect of the crude extracts against *Ae. albopictus* females ovipositing in the open field was evaluated.

**Abstract:**

As there has been no scientific evidence of the bioactivity of *Sambucus ebulus* (Adoxaceae) extracts against insects, we chemically characterized *S. ebulus* leaves and flowers extracted in methanol and water. The crude extracts, phenolic compounds, and amino acids isolated were tested as larvicides against the fourth-instar larvae of *Aedes albopictus* and *Culex pipiens* (Diptera: Culicidae). To understand their mode of action, we evaluated the in vitro acetylcholinesterase (AChE) inhibitor effect of the crude extracts on the two mosquito larvae through a colorimetric method. Furthermore, the deterrent effect of the crude extracts against ovipositing *Ae. albopictus* females was assessed in the open field. Twelve phenylpropanoids and fourteen amino acids were detected in the extracts, with a prevalence of hydroxycinnamic acids and nonaromatic amino acids. The most toxic compound to *Ae. albopictus* larvae after 24 h was gallic acid, followed by the crude *S. ebulus* leaf extract; on *Cx. pipiens*, it was the crude flower extract. The AChE test showed higher inhibition on both mosquito species exerted by the leaf extract if compared to the flower extract, and it also deterred oviposition by *Ae. albopictus* females starting from the third day. The results indicated that vegetal extracts could effectively help in the integrated vector management of mosquitoes.

## 1. Introduction

*Sambucus ebulus* L. (Adoxaceae), known as the dwarf elder for its moderate height (1–1.5 m), is an herbaceous perennial plant characterized by opposite, pinnate leaves and corymb inflorescences of white or pinkish flowers [1]. It is widespread in woods, uncultivated lands, ditches, damp places, and along country lanes from central and southern Europe to northwest Africa and southwest Asia. In Europe, it blooms from June to August [2]. In most of the countries where the species is wildly spread, it is used in folkloristic medicine against diseases of an inflammatory nature, such as gastrointestinal ailments, oedemas, skin wounds, and rheumatoid arthritis [3,4]. The anti-inflammatory activity is well supported by scientific pieces of evidence [5,6]. Furthermore, the crude extracts in different solvents of various *S. ebulus* organs (i.e., leaves, stems, flowers, fruits, and rhizomes) or their pure compounds were successfully applied in vitro as antioxidants [7,8] and against bacteria, fungi [9,10], and cancer cell types [11]. Polyphenols, anthocyanins, flavonoids, glycosides, hydrocarbons, fatty acids, steroids, sugar-binding proteins (lectins), and ribosome-inactivating proteins (ebulins) are the pharmacologically active compounds responsible for all the properties of *S. ebulus* [12,13,14].

*Sambucus ebulus’* leaves and stems preparations are reported in Turkish ethnoveterinary medicine as useful against external parasites [15,16] and, in central Italy, its branches are traditionally hung as an insect repellent [17]. However, to the best of our knowledge, there are no specific studies about the application of dwarf elder extracts in the entomological field.

Among the insect pests of most relevance, we can surely name all the health-threatening mosquitoes as vectors of bacteria, parasites, and viruses. In detail, this work is focused on two cosmopolitan Diptera Culicidae: the Asian tiger mosquito *Aedes albopictus* (Skuse) and the common house mosquito *Culex pipiens* L. *Aedes albopictus*, native to tropical and subtropical forests of Southeast Asia, was first reported outside Asia in Albania, Texas, and Brazil in 1979, 1985, and 1986, respectively [18,19,20]. Afterward, it shortly established itself in other American countries, West and Central Africa, the Pacific, Indian Ocean, and Caribbean islands, Australia, and New Zealand. Indeed, it is considered as one of fastest spreading invasive mosquito species [21,22,23,24,25,26]. *Ae. albopictus* is one of the main vectors of arboviral diseases such as dengue, chikungunya, and Zika [27]. *Culex pipiens* is native to North Africa, but it is currently spread across this whole continent, North and South America, Europe, the Middle East, and Asia. Among the transmitted viruses by the *Cx. pipiens* complex, we can list West Nile, Japanese encephalitis, Saint Louis encephalitis, and Rift Valley fever, also transmitted by *Ae. albopictus* [28]. Such infections cause more than 700,000 deaths each year [29], with higher incidence in the austral hemisphere, especially in South American and South African countries [30]. The control of mosquito populations is currently achieved by repeatedly turning to chemical insecticides such as pyrethroids and organophosphates against adults and insect growth regulators on larvae, although there have been several reported cases of resistance [31] and non-target mortality [32]. Anyhow, the management of such menaces is as fundamental as the research for more sustainable, affordable, and targeted strategies to avoid environmental pollution and risks to human and animal health. Following the report of the Italian Istituto Superiore di Sanità (ISS) [33], mosquito control measures must be based on an IVM (Integrated Vectors Management) that includes the search for and removal of foci of larvae development and environmental remediation or the use of larvicidal products in foci that cannot be removed or remediated and of adulticide products in emergency situations. It is emphasised that the European legislation Water Framework Directive 2000/60/EC [34] obliges member states to achieve the good chemical and ecological status of surface and groundwater bodies and, above all, obliges them to enforce the principle of “non-deterioration” of a water body usually colonised by *Ae. albopictus* and *Cx. pipiens*. Thus, for larvicidal treatments, environmentally friendly substances are strongly recommended.

With this aim, the goal we set for ourselves was to perform a screening to verify the possible usefulness of the dwarf elder in the control of mosquito populations. Therefore, we extracted and chemically characterized (through chromatographic techniques) leaves and flowers of *S. ebulus* components (i.e., phenylpropanoids (in methanol) and amino acids (in water)). The two crude extracts and some of the phenolic compounds (chlorogenic, gallic, hydroxybenzoic, and rosmarinic acids) and amino acids (arginine, aspartate, glutamic acid, histidine, serine, and threonine) isolated were tested as toxic agents for *Ae. albopictus* and *Cx. pipiens* fourth-instar larvae under laboratory conditions. To better understand their mode of action, we evaluated the in vitro acetylcholinesterase (AChE) inhibitor effect of the crude extracts on the two mosquito larvae through a colorimetric method. Furthermore, the deterrent effect of the crude extracts against ovipositing *Ae. albopictus* females was assessed in the open field.

## 2. Materials and Methods

### 2.1. Sambucus ebulus Vegetal Material

*Sambucus ebulus* leaves and flowers were harvested along the riverbanks of the Arno River in Pisa (Tuscany, Italy), where the plant grows luxuriantly, during July 2021. Plant material was frozen immediately in liquid nitrogen and stored at −80 °C after harvest.

The diagram of the experimental design is given in Figure 1.

### 2.2. Preparation of Leaf and Flower Extracts

A 100 mg fresh weight (FW) portion of *S. ebulus* leaves or flowers was homogenized in a mortar with 1 mL of 100% HPLC-grade methanol for phenylpropanoid extraction, while a 50 mg FW portion of leaf or flower tissues was mixed in HPLC water for amino acid extraction. Samples were then incubated overnight in the dark at 4 °C [35]. Extracts were centrifuged for 20 min at 16,000× *g* at 4 °C, and the respective supernatants were filtered through 0.2 μm Ministart SRT 15 filters (Sigma-Aldrich, Milan, Italy) and preserved in test tubes at −20 °C until chromatographic analyses. Both crude extracts were also tested in entomological bioassays against mosquitoes.

### 2.3. Determination and Identification of Phenylpropanoids and Amino Acids

Phenylpropanoids were determined by Ultra-High Performance Liquid Chromatography (UHPLC) using a Dionex UltiMate 3000 system (Thermo Scientific, Waltham, MA, USA) equipped with a reverse-phase Agilent column (ZORBAX Eclipse plus C18, 5 μm particle size, 4.6 mm internal diameter × 150 mm length; Agilent Technologies, Inc., Santa Clara, CA, USA) kept at 30 °C and a Dionex UVD 17U detector (Thermo Scientific). The analytical conditions for the UHPLC determination were as follows: 100% solvent A ((water/methanol/acetic acid, 75:20:5 (*v*/*v*/*v*)) eluted for 1 min, a 30 min linear gradient to 100% solvent B (water/methanol/acetic acid, 50:45:5, (*v*/*v*/*v*)), a 5 min linear gradient to 100% solvent A, and, finally, 5 min of 100% solvent A. The flow rate was 1.0 mL min^−1^ and the peaks were acquired and quantified at 280 and 590 nm [36]. The standard curves were obtained by plotting increasing concentrations of standard methanolic solutions as a function of peak area in UHPLC chromatograms. Apigenin, catechin, chlorogenic acid, dihydroxybenzoic acid, gallic acid, hydroxybenzoic acid, kaempferol, luteolin, neochlorogenic acid, rosmarinic acid, rutin, and sinapic acid were determined and quantified in each extract. The raw chromatographic data were processed by Chromeleon Chromatography Management System software, version 7.2.10-2019 (Thermo Scientific).

The amino acid contents were analyzed by the same UHPLC system above reported and equipped with an Agilent column (Zorbax Eclipse AAA column, 5 μm particle size, 4.6 mm internal diameter × 150 mm length; Agilent, Milan, Italy) kept at 40 °C. A pre-column derivatization was performed by neutralizing samples in 0.4 M borate buffer (pH 10.2) to ensure that the amino terminus of each amino acid was neutralized. Primary amines were then reacted with phthalaldehyde and secondary amines (such as proline) were reacted with 9-fluorenylmethylchloroformate. The analytical conditions for the UHPLC determination were as follows: 100% solvent A (demineralized water 40 mM sodium phosphate dibasic buffer (pH 7.8)) for the first 18 min followed by a 1 min linear gradient to 57% solvent B (acetonitrile:methanol:water, 45:45:10 *v*/*v*), progressively set to reach 100% solvent B in 4 min, followed by a 5 min linear gradient to 100% solvent A. The flow rate was 2 mL min^−1^. Primary amino acids were detected with a fluorescence detector (FLD-3400 RS; Thermo Scientific) at 340 and 450 nm (excitation and emission fluorescence), whereas secondary amino acids were detected with the UV detector above reported at 262 nm [37]. The standard curves were obtained by plotting increasing concentrations of standard water solutions as a function of peak area in UHPLC chromatograms. Alanine, arginine, aspartic acid, glycine, glutamic acid, glutamine, histidine, isoleucine, leucine, phenylalanine, serine, threonine, and valine were determined and quantified in each extract using the software reported above. Measurements were performed on three different extracts belonging to three biological replicates; two instrumental replicates were carried out for each extract.

### 2.4. Collection and Rearing of Mosquito Eggs and Larvae

Over the Summer and early Autumn of 2021, we collected *Ae. albopictus* eggs by placing ovitraps consisting of black pots provided with four brown masonite strips (ovistrips) in the garden of the entomology laboratory at the Department of Agriculture, Food and Environment (DAFE) of the University of Pisa (Italy), as seen in Müller et al. [38]. The ovistrips were picked up and replaced every other day, and eggs were left to hatch in a plastic basin (35 × 27 × 13 cm) filled with aged tap water under laboratory conditions (25 ± 1 °C, 45 ± 5% RH, and natural photoperiod). Tap water was aged for 24 h to dissipate the chlorine. *Ae. albopictus* larvae were then reared using dry cat food as nourishment. At the same time, we collected *Cx. pipiens* egg rafts from concrete tanks (80 × 60 × 45 cm) placed in the same garden of the DAFE and let them hatch in a separate plastic basin filled with aged tap water and provided with dry cat food under the same laboratory conditions. The confirmation of the identification of the two mosquito species was performed on larvae thanks to the pictorial keys provided by Andreadis et al. [39]. The fourth-instar larvae of the two species, both obtained from the eggs collected outdoors, were then employed in the larvicidal tests.

### 2.5. Toxicity Tests on Aedes albopictus and Culex pipiens Larvae

The larvicidal activity of the leaf and flower *S. ebulus* extracts and the selected phenolic and amino acid compounds was evaluated on *Ae. albopictus* and *Cx. pipiens* following the specific 2005 WHO protocol [40] with minor modifications.

Crude extracts of flowers and leaves at increasing concentrations ranging from 2.0 to 4.0 mg mL^−1^ were diluted in aged tap water. Likewise, single phenolic compounds at increasing concentrations ranging from 1.0 to 5.0 mg mL^−1^ and single amino acids ranging from 0.50 to 2.5 mg mL^−1^ were diluted in aged tap water. All the toxicity tests were preceded by preliminary assessments to select the most suitable concentrations to obtain mortality values >0% and <99%. Groups of ten newly emerged fourth-instar larvae (0–24 h old) of both species were separately put in beakers containing 250 mL of solution of the *S. ebulus* extracts or singular compounds at the different concentrations indicated. As a negative control for vegetal extracts and phenylpropanoids, ten larvae were put in 250 mL of a solution of aged tap water and methanol at the concentration of 50 µL mL^−1^ (same quantity of methanol in the extracts and phenylpropanoid solutions at the highest concentration tested). For amino acids, controls were performed in 250 mL of aged tap water. For each treatment, control, and mosquito species, we performed four replicates. During the test, no food was given to the larvae according to Conti et al. [41]. Larval mortality was recorded after the 24 h of permanence in the treatments.

### 2.6. Acetylcholinesterase Extraction and Inhibition Effect on Aedes albopictus and Culex pipiens Larvae

Since one of the most frequently reported modes of action on insects exerted by plant compounds, especially phenols, is the inhibition of AChE, we extracted this enzyme and evaluated its inhibition according to Bedini et al. [42,43] with minor modifications. We exclusively tested the effect of *S. ebulus* leaf and flower extracts on *Ae. albopictus* and *Cx. pipiens* larvae. An aliquot consisting of 300 mg of fourth-instar larvae of both mosquito species was individually homogenized in 4 mL of a buffer (10 mM Tris-HCl, pH 8.0) containing 0.5% (*v*/*v*) Triton X-100 and 20 mM NaCl. The homogenate was centrifuged at 17,000× *g* at 4 °C for 15 min, and the supernatant containing the AChE was filtered through glass wool to remove any excess lipids. The total protein content was quantified by the Protein Assay Kit II (Bio-Rad Laboratories, Inc., Hercules, CA, USA), and the enzymatic extract was used for the following assay.

Inhibition of AChE was determined by the classical colorimetric method of Ellman et al. [44], using acetylthiocholine (ACh) as substrate. The protein content of the AChE extract was diluted to 0.1 mg mL^−1^, and the reaction mixture consisted of 500 μL of diluted AChE extract (which contained 0.05 mg of protein) and 50 μL of leaf or flower extracts for each concentration (2, 5, 25, 50, 100, 125, 250, and 500 µL mL^−1^) dissolved in 100% (*v*/*v*) methanol. Pure methanol was added to reach the final volume of 1 mL in the cuvette. Negative controls were prepared by adding methanol at the same concentrations but without the extracts. The tube was set in an incubator at 25 °C for 5 min before adding 100 μL of 0.01 M5, 5′-dithiobis-(2-nitrobenzoic acid) (DTNB; dissolved in phosphate buffer pH 7.0) and 2.4 mL of phosphate buffer (pH 8.0). The mixture was gently stirred and incubated for 10 min at 25 °C before adding 40 μL of 75 mM ACh (dissolved in 0.1 M phosphate buffer pH 8.0) and then incubated again for 20 min at 25 °C.

The AChE activity was measured using an Ultrospec2100 Pro spectrophotometer (GE Healthcare Ltd., Amersham, UK) at 25 °C, by measuring the increase in absorbance at 412 nm. The inhibition percentage of AChE activity was calculated as follows:AChE inhibition % = (1 − SAT/SAC) × 100
where SAT is the specific activity of the enzyme in the treated group and SAC is the specific activity of the enzyme in the control group.

The residual percentage of AChE activity was calculated as:Residual percentage % = (SAT/SAC) × 100

For each *S. ebulus* extract, concentration, and mosquito species, we performed three replicates.

### 2.7. Oviposition Deterrence towards Aedes albopictus in the Open Field

As the reproductive ethology of *Cx. pipiens* is completely different from that of *Ae. albopictus*, oviposition deterrence tests were only carried out on the latter species. Actually, *Cx. pipiens* lays its eggs in large natural bodies of water where it is not possible to test the extracts, whereas *Ae. albopictus* oviposits in small containers, such as the ovitraps previously described. Therefore, the oviposition deterrence exerted by the *S. ebulus* leaf and flower extracts was evaluated only on *Ae. albopictus* gravid females in the open field, according to the methodology described by Benelli et al. [45]. The trials were carried out in the garden of the entomology laboratory at the DAFE from June to August 2022. The *S. ebulus* extracts were separately tested at the concentration of 2.0 mg mL^−1^ in aged tap water. Such concentration was chosen to be slightly below the LC_50_ for both extracts. For each treatment, a black pot was filled with 500 mL of the solution; as a negative control, we used 500 mL of aged tap water. The three pots (two treatments, meaning the two extracts, and one control) were arranged in lines and separated by 50 cm among each other and 80 cm away from the other groups of pots. The position of the pots was alternated among the five replicates to avoid oviposition bias due to positional effects [45]. In each pot belonging to the treated or control group, we placed one brown masonite strip (200 × 25 mm) (ovistrip) to allow *Ae. albopictus* females to lay eggs on. The number of eggs on the ovistrips was counted five times (i.e., after 1, 2, 3, 4, and 7 days) under a stereomicroscope (Leica EZ4, Leica Biosystems Italia, Buccinasco, Italy). All the ovistrips were collected and replaced in correspondence with each egg count. The data were analysed as Oviposition Activity Index (OAI), using the following formula by Kramer and Mulla [46]:OAI = (NT − NC)/(NT + NC)
where NT is the total number of eggs in the treated pots and NC is the total number of eggs in the control pots. Positive OAI values indicate that more eggs were laid in the treated pots than in the control ones, so the tested extract was attractive. Conversely, negative OAI values indicate that more eggs were laid in the control pots than in the treated ones, so the tested extract was repellent [46].

### 2.8. Data Analyses

The effects of vegetal tissue (leaf vs. flower) on biochemical parameters (i.e., phenylpropanoids and amino acids) were analysed by the Student’s *t*-test.

The median lethal concentration (LC_50_) of the *S. ebulus* extracts and phenolic compounds against *Ae. albopictus* and *Cx. pipiens* larvae was calculated by Log-probit regression [47]. The fitness of the probit model (PROBIT(p) = Intercept + BX; where PROBIT(p) is the cumulative probability estimates, B is the slope of the model, and X is the extract concentration transformed using the base 10 logarithm (covariate)) was tested through the Pearson Goodness-of-Fit Test. A heterogeneity factor was used in the calculation of confidence limits when the significance level was less than 0.150. Differences between LC_50_ values were assessed by Relative Median Potency (RMP) estimates, and they were considered significant if the RMP 95% confidence interval did not include 1. Differences between mosquito species susceptibility and among *S. ebulus* extracts and phenolic compounds toxicity were tested by one-way between-group univariate analysis of covariance (ANCOVA), with the chemical compound (extract or phenol) as a fixed factor and the concentration applied as a covariate to control its effects in the model. The mean response for each factor (chemical compound), adjusted for the concentration, was reported as estimated marginal (EM) means, and significant differences among them were determined by post-hoc comparisons using Bonferroni corrections for multiple comparisons. Statistical analyses were performed by the SPSS 22.0 software (IBM SPSS Statistics, Armonk, North Castle, NY, USA).

Data of AChE activities were arcsine transformed before analyses (after checking with the Shapiro–Wilk test the normal distribution of data) and then subjected to one-way analysis of variance (ANOVA). Significant differences between extract concentrations (within each species and plant organ) were determined by Fisher’s least significant difference (LSD) post-hoc test (*p* ≤ 0.05). ANOVA statistical analyses were performed using GraphPad (GraphPad, La Jolla, CA, USA).

Oviposition Activity Index values ≥ 0.3 were considered attractive, and values ≤ −0.3 were repellent according to Kramer and Mulla [46].

## 3. Results

### 3.1. Chemical Composition

Twelve phenylpropanoids were detected in methanol extracts from leaf and flower materials: seven phenolic compounds (four hydroxycinnamic acids and three hydroxybenzoic acids), four flavonoids (two flavones, one flavonol glycoside, and one flavonol), and one catechin (Table 1). Overall, hydroxycinnamic acids represented the most abundant chemical class of phenylpropanoids in leaf and flower materials (in particular, sinapic acid exhibited the highest amount in both organs), followed by hydroxybenzoic acids (in particular, dihydroxybenzoic acid exhibited the highest amounts in leaf, whereas gallic acid was the most abundant in flower material) and flavonol (i.e., kaempferol in both organs). Conversely, flavones (apigenin and luteolin), flavonol glycoside (rutin), and catechin were far less represented (in the flower material).

Fourteen amino acids were detected in water extracts from leaf and flower materials: eleven nonaromatic and three aromatic amino acids (Table 2). Among the eleven nonaromatic amino acids identified, glutamic acid and valine exhibited the highest amounts in leaf and flower extracts, respectively. Conversely, aromatic amino acids (phenylalanine, tyrosine, and serine) were far less represented (in particular, phenylalanine and tyrosine were detected only in the flower material).

From the preliminary screening on all detected phenylpropanoids (see Table 1), only two hydroxycinnamic acids (chlorogenic and rosmarinic) and two hydroxybenzoic acids (gallic and hydroxybenzoic) were selected to be singularly tested on the mosquito species. Since the preliminary tests on amino acids were unclear, we continued our investigation on five nonaromatic (arginine, aspartate, glutamic acid, histidine, and threonine) and one aromatic (serine) amino acids. Such amino acids were selected because no significant differences were observed in terms of chemical abundance in leaf and flower material (see Table 2).

### 3.2. Larvicidal Activity of Sambucus ebulus Leaf and Flower Extracts and Their Phenolic Compounds

From our screening, the leaf and flower extracts were found to have a toxic effect on the two mosquito larvae after 24 h. According to ANCOVA outputs, the larvicidal activity was dependent on the mosquito species (*F*_1,46_ = 17.177; *p* < 0.001) and on the extract concentration (*F*_1,46_ = 191.192; *p* < 0.001) but not on the plant organ (*F*_1,46_ = 0.028; *p* = 0.868). Regarding the leaf extract, the most susceptible species to it was *Ae. albopictus* with an LC_50_ value of 2.55 mg mL^−1^, whereas the LC_50_ value for *Cx. pipiens* was 3.18 mg mL^−1^ (Table 3). The RMP analysis showed that the difference in the susceptibility between the two species was significant (*Ae. albopictus* vs. *Cx. pipiens* RMP = 1.244; CI: 1.123–1.426). Conversely, *Cx. pipiens* larvae were the more susceptible to the extract from *S. ebulus* flower, with an LC_50_ value of 2.77 mg mL^−1^ (Table 3). The RMP analysis showed that the difference in the susceptibility between the two species was significant (*Ae. albopictus* vs. *Cx. pipiens* RMP = 1.069; CI: 0.814–1.468).

The selected phenylpropanoids from *S. ebulus* showed toxic activity against the mosquito larvae, with LC_50_ values ranging from 2.37 to 3.61 mg mL^−1^ for *Ae. albopictus* (Table 4) and from 3.55 to 3.91 mg mL^−1^ for *Cx. pipiens* (Table 4). Anyhow, both for *Ae. albopictus* and *Cx. pipiens* larvae, the RMP analyses indicated no significant differences among the phenolic compounds’ toxicity.

In accordance with the probit analyses, the ANCOVA indicated no statistically significant differences between species (*F*_1,87_ = 1.623, *p* = 0.206) nor among the selected phenylpropanoids of *S. ebulus* (*F*_3,87_ = 1.191, *p* = 0.318) but a significant interaction between species and compounds (*F*_2,32_ = 13.559, *p* < 0.001). In detail, there was a significant effect of the concentration of the phenolic compounds on *Cx. pipiens* larvae mortality (*F*_2,32_ = 9.666, *p* < 0.001) and, for *Ae. albopictus*, a significant effect of both concentration and the phenol on larval mortality (*F*_1,43_ = 211.824, *p* < 0.001 and *F*_3,43_ = 4.338, *p* = 0.009, respectively).

Conversely, all tested amino acids were unable to exert toxicity on the larvae of the two mosquito species at the applied concentrations (mortality was 4.7% only at the highest concentration tested of glutamic acid, namely 2.5 mg mL^−1^, and null for the others). We decided not to test any further concentrations, as the amino acids’ toxicity was not measurable in correspondence with values that almost represented the LC_50_ for the other compounds and extracts involved in the study.

Overall, the most toxic compound on *Ae. albopictus* larvae was gallic acid (LC_50_ = 2.37 mg mL^−1^), strictly followed by the *S. ebulus* leaf extract (LC_50_ = 2.55 mg mL^−1^); on *Cx. pipiens,* it was the flower extract (LC_50_ = 2.77 mg mL^−1^).

### 3.3. Acetylcholinesterase Inhibition in Aedes albopictus and Culex pipiens Larvae

The *S. ebulus* leaf and flower extracts inhibited the in vitro activities of AChE isolated from both *Ae. albopictus* and *Cx. pipiens* larvae (Figure 2). Overall, the leaf extract was more effective in AChE inhibition when compared to the flower one, and the effect was consistent for both the mosquito species under investigation. In particular, the residual AChE activities declined starting from 25 µL mL^−1^ (for *Cx. pipiens*) and 50 µL mL^−1^ (for *Ae. albopictus*) and then paralleled the severity of the treatment. With the flower extract, the inhibitory effect was milder, as only the two highest doses (250 and 500 µL mL^−1^) resulted in dramatic losses of AChE catalytic efficiency tested in vitro.

### 3.4. Repellency towards Aedes albopictus Gravid Females in the Open Field

From our screening in the open field, the leaf extract deterred the oviposition by *Ae. albopictus* starting from the third day after exposure to the treatments. On the contrary, a slightly attractive effect was observed for the flower extract, apart from the fourth day, when the observed effect was a reduction in oviposition. However, the fluctuation of this OAI value around the average (−0.32 ± 0.25) indicates that, in a large proportion of cases, the activity was not significant (Table 5).

## 4. Discussion

Many phytochemical compounds were isolated and identified in different organs of *S. ebulus,* such as phenylpropanoids (i.e., caffeic acid derivatives and flavonoids), amino acids (i.e., ebulitins), steroids, and tannins [12]. The main constituents with therapeutic properties of this plant are the phenolic compounds, found in great numbers in *S. ebulus* leaves and flowers, which are associated with organoleptic, nutritional, and antioxidant properties [13]. Some of them are common in other medicinal plants too (e.g., *Hypericum perforatum* L.–Hypericaceae, *Melissa officinalis* L.–Lamiaceae, *Salvia officinalis* L.–Lamiaceae, as seen in Pellegrini et al. [35], Döring et al. [48], and Marchica et al. [36]) and are responsible for the antioxidant effects. In addition, ebulitin, ebulin 1, flavonoid, anthocyanin, and other components were isolated from *S. ebulus* and identified as other active ingredients with biological and pharmacological activities [12,14]. In our work, we identified twelve phenylpropanoids through UHPLC, and all have been already reported in *S. ebulus* leaf extracts through similar experimental methods and often detected also in dwarf elder roots and fruits [49,50]. Among them, neochlorogenic acid, rosmarinic acid, gallic acid, kampferol, and apigenin were also recorded in dwarf elder flowers extracts [51]. To the best of our knowledge, although the phytochemical composition of *S. ebulus* fruits is well characterized with a detailed list of the amino acidic component [12], the general amino acid profile of *S. ebulus* leaves and flowers has not been deeply investigated. However, from the accurate description in terms of molecular and biological properties of ebulitins, a specific class of proteins extracted from dwarf elder leaves [52], we found that their structure made of sixteen amino acids matches the leaf profile of *S. ebulus* crude extract (in water) resulting from the analyses carried out in this work for fourteen (only cysteine and methionine do not coincide).

As stated before, this is the first attempt to test the dwarf elder in the entomological field, despite its folkloristic use against external parasites in livestock [15,16] as an insect repellent [17]. To the best of our knowledge, there are no studies about the application of *S. ebulus*, nor even plants belonging to the same genus or family (Adoxaceae), to control insect pests. There are, instead, several recommendations to use methanolic extracts of plant tissues to control health-threatening Culicidae populations thanks to the acute toxicity exerted on the treated larvae (or pupae) and adverse effects on their following development and emergence [53,54,55,56]. Although each vegetal extract has a different and peculiar composition compared to the ones here used, as they are derived from diverse plants and organs, we present in Table 6 the results obtained from our screening by comparing our LC_50_ values to those reported by other authors. We also need to consider that the sensitivity is almost species-specific, as demonstrated by our trials and others [53,57].

About the mode of action, the bioactive compounds contained in the vegetal extracts, especially phenols, can enter into contact with the external surface of the larval body or penetrate it through the mouthparts or spiracular openings [54].

Once penetrated, the bioactive compounds can impact the nervous system and alter the functioning of several enzymes by increasing the ATPase activity and inhibiting the cytochrome P-450 monooxygenase, carboxylesterase, and AChE [62]. AChE is the enzyme catalysing the breakdown of the neurotransmitter acetylcholine and is found in neuro-neuronal and neuromuscular junctions in insects and mammals, but insect AChE differs from mammal AChE by a cysteine residue making it a crucial insect-selective target [63]. In the case here reported, using the colorimetric method of Ellman et al. [44], especially the *S. ebulus* leaf extract significantly inhibited the in vitro activity of the AChE isolated from both *Cx. pipiens* and *Ae. albopictus* individuals from, respectively, 25 and 50 µL mL^−1^. Following a similar protocol, Demouche et al. [56] indicated a percentage of AChE inhibitory activity corresponding to 30.98 ± 2.97% in *Cx. pipiens* fourth-instar larvae into contact with a methanolic extract of dried aerial parts of *Cotula cinerea* Delile (Asteraceae) at 0.90 mg mL^−1^. Moreover, through a semi-microtechnique, El Hadidy et al. [62] assessed that the AChE of *Cx. pipiens* third-instar larvae was reduced by, respectively, 57.86, 40.97, and 15.95% when treated with hexane extracts from the dried flowers, leaves, and stems of *Ageratum houstonianum* Mill. (Asteraceae) at concentrations of 259.79, 266.85, and 306.86 ppm.

At the alimentary canal level, some extracts can damage the digestive cells and degrade the microvilli and peritrophic membrane, causing the release of the gut content, disruption in the absorption of nutrients, and imbalanced homeostasis. Such histopathological alterations were shown in *Cx. pipiens* third-instar larvae treated with a *Lepidium sativum* L. (Brassicaceae) ethyl acetate extract [55]. However, further research is needed to verify whether this is an additional mode of action that *S. ebulus* extracts can have on the two mosquito species involved in the study here reported.

In addition to the acute toxicity and physiological and morphological alterations in larvae, some vegetal extracts can similarly affect pupae [53,56] and cause weakness and fly inability in the adults emerged from specimens that survived treatments [54].

As for the tests on ovipositing mosquito females, in our study, the *S. ebulus* leaf extract at a concentration of 2.0 mg mL^−1^ showed an oviposition deterrent action from the third to the seventh day after exposure to the treatment (OAI values of −0.67, −0.68, and −0.43). Similarly, the methanolic extract of neem cake (from the seeds of *Azadirachta indica* A. Juss.–Meliaceae) at 100 ppm against *Ae. albopictus* females gave an OAI of −0.62 after 7 days [45]. At about 0.5 mg mL^−1^, the OAI values reported for methanolic extracts from the dried leaves of *Andrographis paniculata* (Burm. fil.) Nees. (Acanthaceae), *Eclipta prostrata* (L.) L. (Asteraceae), and *Tagetes erecta* L. (Asteraceae) were, respectively, −0.84, −0.82, and −0.87 after 24 h on *Anopheles subpictus* Grassi [64]. On the same mosquito species, the OAI values for the methanolic extracts of *Aegle marmelos* (L.) Corrêa (Rutaceae), *Andrographis lineata* Nees (Acanthaceae), and *Cocculus hirsutus* (L.) W. Theob. (Menispermaceae) dried leaves were, respectively, −0.91, −0.86, and −0.86 after 24 h [65].

Some authors used organic solvents other than methanol to extract the bioactive compounds from different plants or dried powders obtained from vegetal tissues as repellents or mosquitocidals. It would be interesting to investigate the physiological and morphological changes in mosquitoes treated with similar *S. ebulus* products to better understand their effectiveness and possible mode of action.

## 5. Conclusions

The control of mosquito-borne diseases vectors is of unquestionable importance. Considering that their eradication is not realistically possible, limiting their presence and, consequently, the transmission of bacteria, parasites, and viruses, should be the main goal.

IVM of *Ae. albopictus* and *Cx. pipiens* on several fronts, involving not only chemical insecticides in emergency situations but also biological control agents or botanicals, could be a promising strategy. Overall, the *S. ebulus* leaf and flower extracts and their selected phenolic compounds proved to produce a toxic action on *Ae. albopictus* and *Cx. pipiens* larvae. Especially in our case, the crude dwarf elder extracts were roughly as toxic as their main phenolic compounds; therefore, the supplementary steps for their isolation could be avoided.

In conclusion, in addition to being a valuable way to exploit wild plants, the application of vegetal extracts is more affordable, ecological, and safer compared to chemical insecticides and could effectively help in the IVM of health-threatening mosquito populations. However, in order for vegetal extracts to be recommended and used by official bodies (ISS), the challenge has to be taken up by the industry, which will have to formulate and dose them in the most appropriate way.

## Figures and Tables

**Figure 1 insects-15-00482-f001:**
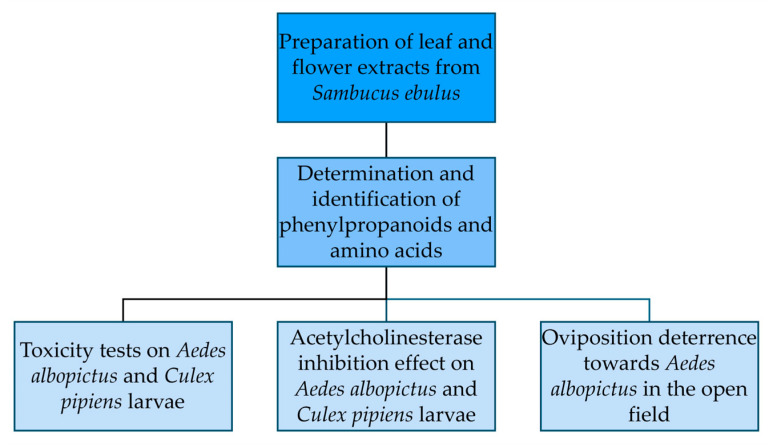
Diagram of the experimental design, starting from the extraction of *Sambucus ebulus* leaves and flowers and continuing with the chemical determination and identification of the phenylpropanoids and amino acids contained and entomological bioassays (larval toxicity, acetylcholinesterase inhibition effect, and oviposition deterrence).

**Figure 2 insects-15-00482-f002:**
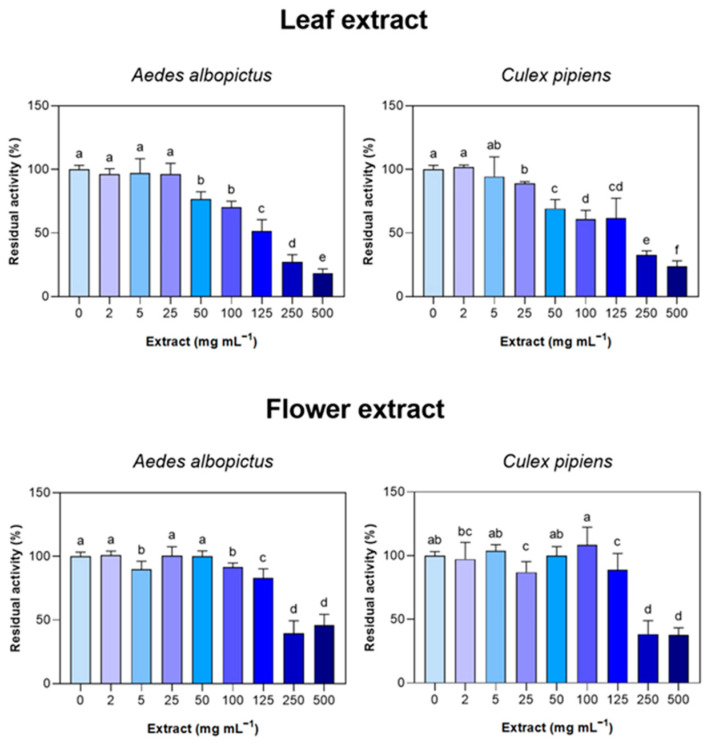
In vitro residual percentage of AChE activity on *Aedes albopictus* and *Culex pipiens* after the application of the leaf and flower extracts of *Sambucus ebulus*. Data are given as means (*n* = 3) ± standard error. Different letters are significantly different after one-way ANOVA followed by Fisher’s LSD post-hoc test (*p* ≤ 0.05).

**Table 1 insects-15-00482-t001:** Hydroxycinnamic acids, hydroxybenzoic acids, flavones, flavonol glycoside, flavonol, and catechin (mg g^−1^ fresh weight) in *Sambucus ebulus* leaves and flowers. Data are shown as means ± standard errors (*n* = 3 biological replicates for each tissue extract). For each compound, significant differences between the two plant organs are for *** *p* ≤ 0.001; * *p* ≤ 0.05; ns > 0.05.

	Leaf Extract [mg g^−1^ FW]	Flower Extract [mg g^−1^ FW]	*p*
Hydroxycinnamic acids			
Chlorogenic acid	0.329 ± 0.03	0.325 ± 0.02	ns
Neochlorogenic acid	0.8 ± 0.01	0.06 ± 0.002	*
Sinapic acid	1.86 ± 0.2	2.53 ± 0.3	ns
Rosmarinic acid	0.39 ± 0.02	0.34 ± 0.01	ns
Hydroxybenzoic acids			
Dihydroxybenzoic acid	0.35 ± 0.01	0.11 ± 0.02	***
Gallic acid	0.26 ± 0.01	0.19 ± 0.02	ns
Hydroxybenzoic acid	0.033 ± 0.001	0.03 ± 0.001	ns
Flavones			
Apigenin	0.14 ± 0.01	0.2 ± 0.01	***
Luteolin	0.5 ± 0.01	0.49 ± 0.05	ns
Flavonol glycoside			
Rutin	1.06 ± 0.11	0.27 ± 0.02	*
Flavonol			
Kaempferol	0.63 ± 0.05	0.88 ± 0.02	ns
Catechin	0.64 ± 0.01	0.26 ± 0.02	ns

**Table 2 insects-15-00482-t002:** Nonaromatic and aromatic amino acids (mg g^−1^ fresh weight) in *Sambucus ebulus* leaves and flowers. Data are shown as means ± standard errors (*n* = 3 biological replicates for each tissue extract). For each compound, significant differences between the two plant organs are for ** *p* ≤ 0.01; * *p* ≤ 0.05; ns > 0.05. nd: not detected.

	Leaf Extract [mg g^−1^ FW]	Flower Extract [mg g^−1^ FW]	*p*
Nonaromatic			
Alanine	0.2 ± 0.03	0.7 ± 0.01	**
Arginine	0.17 ± 0.01	0.17 ± 0.04	ns
Aspartate	0.23 ± 0.02	0.24 ± 0.01	ns
Glutamic acid	0.81 ± 0.03	1.0 ± 0.2	ns
Glutamine	0.33 ± 0.02	1.26 ± 0.2	*
Glycine	nd	1.7 ± 0.2	
Histidine	0.36 ± 0.01	0.56 ± 0.03	ns
Isoleucine	nd	0.42 ± 0.03	
Leucine	nd	0.37 ± 0.03	
Threonine	0.17 ± 0.03	0.14 ± 0.02	ns
Valine	nd	1.76 ± 0.07	
Aromatic			
Phenylalanine	nd	0.21 ± 0.03	
Serine	2.0 ± 0.3	1.47 ± 0.1	ns
Tyrosine	nd	0.45 ± 0.02	

**Table 3 insects-15-00482-t003:** Toxicity of the extracts from *Sambucus ebulus* leaf and flower against *Aedes albopictus* and *Culex pipiens* larvae after 24 h.

Species	LC_50_	95% CI	Intercept ± SE	*p*
Leaf extract				
*Ae. albopictus*	2.55	2.38–2.73	−13.72 ± 1.53	<0.001
*Cx. pipiens*	3.18	2.99–3.41	−12.85 ± 1.48	<0.001
Flower extract				
*Ae. albopictus*	2.96	2.48–3.85	−12.07 ± 1.49	<0.001
*Cx. pipiens*	2.77	2.25–3.39	−11.83 ± 1.50	<0.001

LC_50_ = concentration (mg mL^−1^) of *S. ebulus* extract that kills 50% of the larvae. CI = confidence interval. Intercept = x-intercept of the model regression curve. SE = standard error. *p* = probability that x-intercept = 0. Data are calculated by Probit regression. Model slope for leaf extract = 9.131 ± 1.042; Pearson goodness-of-fit test, *χ*^2^ = 4.493, df = 6, *p* = 0.610. Model slope for flower extract = 8.200 ± 1.033; Pearson goodness-of-fit test, *χ*^2^ = 16.169, df = 5, *p* = 0.006. Since the significance level is less than 0.150, a heterogeneity factor was used in the calculation of confidence limits.

**Table 4 insects-15-00482-t004:** Toxicity of the phenolic compounds extracted from *Sambucus ebulus* against *Aedes albopictus* and *Culex pipiens* larvae after 24 h.

Compounds	LC_50_	95% CI	Intercept ± SE	*p*
*Aedes albopictus*				
Chlorogenic acid	2.90	2.18–3.69	−2.19 ± 0.24	<0.001
Gallic acid	2.37	1.74–3.15	−1.77 ± 0.21	<0.001
Hydroxybenzoic acid	3.61	2.64–4.98	−2.64 ± 0.26	<0.001
Rosmarinic acid	3.05	2.26–4.02	−2.29 ± 0.25	<0.001
*Culex pipiens*				
Chlorogenic acid	3.55	2.86–4.23	−3.92 ± 0.37	<0.001
Gallic acid	3.91	3.17–4.91	−4.22 ± 0.36	<0.001
Hydroxybenzoic acid	3.62	2.87–4.42	−3.98 ± 0.37	<0.001
Rosmarinic acid	3.60	2.92–4.28	−3.96 ± 0.37	<0.001

LC_50_ = concentration of phenolic compounds that kills 50% of the larvae. CI = confidence interval. Intercept = x-intercept of the model regression curve. SE = standard error. *p* = probability that x-intercept = 0. Data are calculated by Probit regression. Model slope for *Ae. albopictus* = 4.733 ± 0.406; Pearson goodness-of-fit test, *χ*^2^ = 46.027, df = 12, *p* < 0.001. Model slope for *Cx. pipiens* = 7.124 ± 0.585; Pearson goodness-of-fit test, *χ*^2^ = 76.271, df = 17, *p* < 0.001. Since the significance level is less than 0.150, a heterogeneity factor was used in the calculation of confidence limits.

**Table 5 insects-15-00482-t005:** Effect of *Sambucus ebulus* leaf and flower extracts against the oviposition of the mosquito *Aedes albopictus* over a week.

Extract	Treatment	Day	Average Number of Eggs	OAI
Leaf	Control	1	19.00 ± 6.66	−0.05 ± 0.13
	Extract		15.33 ± 1.20	
	Control	2	19.33 ± 9.68	−0.19 ± 0.42
	Extract		14.33 ± 5.49	
	Control	3	17.00 ± 9.07	**−0.67 ± 0.33**
	Extract		3.33 ± 3.33	
	Control	4	20.00 ± 6.56	**−0.68 ± 0.09**
	Extract		4.33 ± 2.40	
	Control	7	69.00 ± 36.76	**−0.43 ± 0.21**
	Extract		18.67 ± 8.17	
Flower	Control	1	9.67 ± 6.49	−0.1 ± 0.51
	Extract		4.67 ± 1.67	
	Control	2	10.33 ± 6.89	0.24 ± 0.35
	Extract		26.00 ± 15.82	
	Control	3	11.67 ± 9.17	0.23 ± 0.50
	Extract		12.33 ± 4.91	
	Control	4	50.33 ± 18.17	**−0.32 ± 0.25**
	Extract		25.00 ± 15.10	
	Control	7	60.33 ± 6.17	0.07 ± 0.04
	Extract		68.67 ± 1.86	

Extracts concentration: 2.0 mg mL^−1^. Data are given as means ± standard error (*n* = 5). OAI, oviposition activity index. OAI values ≥ 0.3 indicate that extracts were attractive; OAI values ≤−0.3 indicate that extracts were repellent according to Kramer and Mulla [46] (in bold).

**Table 6 insects-15-00482-t006:** Medial lethal concentrations (LC_50_) exerted by methanolic vegetal extracts towards mosquito (Diptera: Culicidae) larvae.

Plant Species	Organ Extracted	Target Mosquito Species	Larval Instar	LC_50_	Ref.
*Anacardium occidentale* (Anacardiaceae)	Stems	*Aedes albopictus*	Third–fourth	638 mg L^−1^	[57]
*Anacardium occidentale* (Anacardiaceae)	Stems	*Aedes aegypti*	Third–fourth	792 mg L^−1^	[57]
*Catharanthus roseus* (Apocynaceae)	Leaves	*Anopheles stephensi*	Fourth	78.80 mg mL^−1^	[53]
*Catharanthus roseus* (Apocynaceae)	Leaves	*Culex quinquefasciatus*	Fourth	94.20 mg mL^−1^	[53]
*Clausena dentata* (Rutaceae)	Leaves	*Culex quinquefasciatus*	Third	1.87 mg mL^−1^	[58]
*Gluta renghas * (Anacardiaceae)	Stems	*Aedes albopictus*	Third–fourth	240 mg L^−1^	[57]
*Gluta renghas * (Anacardiaceae)	Stems	*Aedes aegypti*	Third–fourth	623 mg L^−1^	[57]
*Leptochloa uniflora * (Poaceae)	Aerial parts	*Culex quinquefasciatus*	Fourth	770.47 mg mL^−1^	[59]
*Mangifera indica* (Anacardiaceae)	Stems	*Aedes albopictus*	Third–fourth	431 mg L^−1^	[57]
*Mangifera indica * (Anacardiaceae)	Stems	*Aedes aegypti*	Third–fourth	582 mg L^−1^	[57]
*Melanochyla fasciculiflora* (Anacardiaceae)	Stems	*Aedes albopictus*	Third–fourth	535.59 mg L^−1^	[57]
*Melanochyla fasciculiflora* (Anacardiaceae)	Stems	*Aedes aegypti*	Third–fourth	696 mg L^−1^	[57]
*Molineria trichocarpa* (Hypoxidaceae)	Aerial parts	*Culex quinquefasciatus*	Fourth	478.51 mg mL^−1^	[59]
*Pancratium triflorum* (Amaryllidaceae)	Aerial parts	*Culex quinquefasciatus*	Fourth	357.47 mg mL^−1^	[59]
*Sambucus ebulus * (Adoxaceae)	Leaves	*Aedes albopictus*	Fourth	2.55 mg mL^−1^	Our results
*Sambucus ebulus * (Adoxaceae)	Flowers	*Culex pipiens*	Fourth	2.77 mg mL^−1^	Our results
*Sambucus ebulus * (Adoxaceae)	Flowers	*Aedes albopictus*	Fourth	2.96 mg mL^−1^	Our results
*Sambucus ebulus * (Adoxaceae)	Leaves	*Culex pipiens*	Fourth	3.18 mg mL^−1^	Our results
*Spermacoce latifolia* (Rubiaceae)	Aerial parts	*Aedes aegypti*	Third	0.62 mg mL^−1^	[60]
*Vernonia cinerea* (Asteraceae)	Leaves	*Culex quinquefasciatus*	Third	2.08 mg mL^−1^	[61]

## Data Availability

Data are available on request from the corresponding author.

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
