# Peer review of "Leaf and Flower Extracts from the Dwarf Elder (Sambucus ebulus): Toxicity and Repellence against Cosmopolitan Mosquito-Borne Diseases Vectors"

_insects, 2024, doi:10.3390/insects15070482_

Round 1

Reviewer 1 Report

Comments and Suggestions for Authors

Abstract section

Lines 26, 29, 31, 32, and 35 italicize the scientific names of both mosquito and plant species.

Lines 35-36. As with other publications using vegetal extracts, there is no roadmap for how to use them within an IVM for mosquitoes. This affirmation needs to be supported by laboratory and semi-field studies to be validated alongside a policy in collaboration with the Ministry of Health at the local and national levels.

Introduction section

This section should include information about the status of Ae. albopictus and Cx. pipiens (confirmed cases, hot spots, insecticide resistance, etc.) and the control program implemented by the MoH in Italy.

The oviposition deterrence experiment should be performed in laboratory conditions.

Materials and method section

A diagram of the experimental design should be included to facilitate understanding of the experiments and analyses.

Line149-164. What was the reference used to determine the different concentrations for the entomological test? Or on what basis did you select these concentrations?

Line 173. italicize Ae. albopictus

How many larvae do you rear per container? How was the amount of food determined for each container? These conditions should be clearly explained (e.g., larvae nutrition should be standardized), especially in this type of bioassay.

Line 181-190. Why did you select 10 larvae per cup, and why did you use 50 mL for the control instead of 250 mL? What was the reference used for? Additionally, considering that this type of bioassay assesses mortality at different time intervals, why did you choose to measure it only at 24 hours?

Line 192. Have you confirmed through biochemical or molecular analyses that both mosquito species (used for bioassays) have mutations responsible for insensitivity to acetylcholinesterase, leading to resistance to insecticides?

Line 229. Do you have information about the seasonality and abundance of Ae. albopictus in the area? Additionally, is the Ministry of Health (MoH) conducting chemical control measures, and if so, how often? It's important to consider any other control measures as well, as they can impact the abundance of mosquitoes and therefore affect the oviposition experiment.

Results section

Figure 1. Poor quality

The oviposition deterrence experiment should be conducted under laboratory conditions as well as open field to minimize external interventions such as routine control methods or variations in climatic conditions.

Discussion section

Line 414-424. It should be started by the main finding. I suggest delete this paragraph or rewrite it.

Line 491-93. While you have observed this effect with other plant species, there is no evidence presented or available to suggest that S. ebulus has the same impact.

I suggest that the discussion address more realistic ways to incorporate plant extracts into and IVM program and the transfer mechanisms to be adopted by the local Ministry of Health. While the information presented is valuable, it currently feels somewhat abstract and disconnected from practical implementation.

It's important for a conclusion to be comprehensive and impactful.

Comments on the Quality of English Language

Minor corrections should be addressed.

Reviewer 2 Report

Comments and Suggestions for Authors

Dear authors,

I believe this in principle is a potentially interesting study on the use dwarf elder extract or specific compounds as mosquito larvicide or repellent. However, I have major concerns with the quality of the data presentation, analysis and interpretation (the latter of which partially might be a language problem, see below), and I don't see the manuscript fit for publication in its current state.

Here are some of my strong concerns about the manuscript:

Throughout the experiments it is not clear which concentrations of the extracts or compounds were tested, in how many replicates, and what presented the data means (averages, means, SD, SEM, p-values etc).

The results descriptions and interpretations in the results text are often hard to comprehend, partly incomprehensible and in parts do not match the numbers presented in the tables.

For tables 3 and 4 it is not clear what the column intercept and P refer to. the numbers don't match the numbers listed in the text. therefore it isn't clear what was analysed here and what these numbers mean, neither the numbers and p-values in the text, nor the numbers in the tables.

Another serious concern is with table 5, where the OAI values listed seem to be completely wrong based on the formula provided in the methods, and the no. of laid eggs shows the opposite of what the OAI value implies and what the authors interpret in the text.

Table and figure captions lack lots of information. All table and figure captions should include following information: on how many biological and technical replicates is the data based?, and how many individuals per replicate were used (if applicable)? What is presented in the tables and figures? i.e. averages, means, medians, SD or SEM?

the methods lack too many details to be able to reproduce the experiments.

Inconsistent use of concentrations throughout the manuscript: mg/mL, µl/mL, µmol/g: should be the same for all the concentrations used to be able to compare the data and connect between the amount found in leaves and the amount used in the assays and in between the assays.

I added many specific comments to the PDF of the manuscript, which I uploaded. However, I did not mark and comment everything I didn't understand and all the English and grammar issues.

Comments on the Quality of English Language

the quality of English language and grammar is not sufficient and would require a thorough English editing and grammar correction for the whole manuscript. Especially in the results section this lack of quality in parts makes the text incomprehensible and / or the statement in the text doesn't match the data shown in the tables. 

Round 2

Reviewer 2 Report

Comments and Suggestions for Authors

Dear authors,

thank you for addressing many of my comments. While some things are clear now, I still have strong concerns with the methods chapter and with Table 5. Please see specifics in my point by point comments below and in the manuscript file. Moreover, several other concerns haven't been sufficiently addressed to improve the quality of the manuscript.

 Reviewer 2 

I believe this in principle is a potentially interesting study on the use dwarf elder extract or specific compounds as mosquito larvicide or repellent. However, I have major concerns with the quality of the data presentation, analysis and interpretation (the latter of which partially might be a language problem, see below), and I don't see the manuscript fit for publication in its current state. 

Here are some of my strong concerns about the manuscript: 

Throughout the experiments it is not clear which concentrations of the extracts or compounds were tested, in how many replicates, and what presented the data means (averages, means, SD, SEM, p-values etc). 

R. We agree with the reviewer that, being a multidisciplinary manuscript, the materials and methods section is quite long and complicated. The concentrations tested in the toxicity trials are reported in lines 190-194, and the repetitions were five. Each oviposition deterrence experimental unit consisted in a control and two different treatments (leaves and flowers extracts). The same control was utilized for the calculation of the OAI of both the treatments. So, the two treatments and the (one) control were replicated five times (see lines 265-269). For data meaning, please see the legend of the tables.

RR: I thank the authors for the additions made to the method section, that solved some of the questions I had. However, I’m still struggling to follow the experiments. Even after reading these sections multiple times and cross-referencing between the methods chapters I can’t figure out what was done in a way that I could repeat the experiments. Therefore, please add the missing information and / or make the descriptions clear.

Specifically, please address the following:

Section: “Preparation of leaf- and flower-methanol extracts for chromatographic analyses”:

If I understand the text correctly then phenylpropanoid extracts were made in 100% methanol and amino acid extracts were made in 100% water (like also described in lines 96-99). However, the title speaks only about methanol extracts. Either the title, is not correct, or I don’t understand the text correctly. In either case, please edit to make it correct and/or make it clearly understandable what was done. 

These extracts would then be analysed by HPLC as described in the follwong chapter, But are they also the “crude extracts” you refer to further below? Then it would be helpful to mention that here. 

Section “Determination and identification of phenylpropanoids and amino acids”

From the methods section above I concluded that the amino acid extracts were in water. But the amino acids standards used for chromatography were in methanol (line 177)? And methanolic solution means 100% methanol? Please specify in the methods

I assume that the HPLC purified compounds were then used for the tests. Specify the solvents and concentrations in which they were obtained from the HPLC for use in the downstream tests. 

Section “Preparation of crude extracts and … compounds”:

The information provided here doesn’t allow to understand how these dilutions were done. First of all: “crude extracts” refers to the extracts from chapter “Preparation of leaf- and flower-methanol extracts for chromatographic analyses”? If yes, as stated above, please add this information to that chapter to make the cross-referencing easier.

If the “crude extracts” are not what is described in that chapter, then the information on how the “crude extracts” were done, is completely missing or not deductible from this chapter. In this case please add in this chapter then how the crude extracts were made.

Second: “methanol solutions of crude extracts were prepared”: how? Provide the details on what was mixed with what in which volumes. The initial crude extract was in what? 100% methanol or 100 % water? Specify!

The Phenolic compounds and amino acids used here I assume come from the HPLC analysis? Please specifiy where they come from or how they were obtained. what was the concentration of these pure compound solutions and what was the solvents

The dilutions of the phenolic compounds were done in methanol (line 192/193). What were the amino acids solutions diluted in?

Section: “Toxicity tests on Aedes albopictus and Culex pipiens larvae”

“As a negative control, ten larvae were put in 250 mL of a solution of water and methanol at the concentration of 5 mg mL -1 (the same methanol concentration of the vegetal extracts).”

My question here is: this control makes only sense if all the dilutions of the leaf and flower extracts and of the single compounds were in 100% methanol, which, as all my questions to the above methods hopefully show, is not clear at all from the descriptions. 

If they dilutions were not all in 100% methanol, then a corresponding control is missing!

I hope, from all these questions you can see how much I struggle to understand the details of what was done. Therefore, please improve this and provide all the requested information in a clearly describing way to meet the scientific standard for reproducibility of experiments.

The results descriptions and interpretations in the results text are often hard to comprehend, partly incomprehensible and in parts do not match the numbers presented in the tables. 

R. We thank the reviewer for the comment, but more details would be helpful in answering the question. However, if the reviewer intends the statistic parameters, please see the replies at lines 284-311. 

RR: this was a general remark on the quality of the results interpretations and descriptions. The specific remarks you addressed below.

For tables 3 and 4 it is not clear what the column intercept and P refer to. the numbers don't match the numbers listed in the text. therefore it isn't clear what was analysed here and what these numbers mean, neither the numbers and p-values in the text, nor the numbers in the tables. 

R. The p-values is one of the coefficients or the probability that, within a given model, the null hypothesis that a particular predictor’s regression coefficient is zero given that the rest of the predictors are in the model. They are based on the Wald test statistics of the predictors.

RR: This isn’t clear to everyone. Please specify in the table caption what “Intercept” and “P” is, the same as you have done for LC50, CI and SE. All table columns should be explained if the column titles are not self-explanatory. Which they aren’t if the reader is not a statistics expert.

Table and figure captions lack lots of information. All table and figure captions should include following information: on how many biological and technical replicates is the data based?, and how many individuals per replicate were used (if applicable)? What is presented in the tables and figures? i.e. averages, means, medians, SD or SEM? 

R. All the tables have a caption but also a legend under reported, where the reviewer can find the detail of what is presented in the table. For replicates, please see lines 264-267. 

RR: this doesn’t answer my point. It is not clear in all tables if the values given are means or medians, for table 1 an d2 it is also not specified if the indicated variances are the standard error or standard deviation. To better be able to judge the presented data it is also helpful to have the information on number of replicates and number or individuals per replicate (it makes it so much easier for the reader to not have to search this latter information in the methods) 

Therefore, I repeat my unaddressed remark: every table of figure caption should contain information on what is presented, i.e. means, medians, SE or SD, and please also add information on no. of replicates and no. of individuals per replicate.

the methods lack too many details to be able to reproduce the experiments. 

Inconsistent use of concentrations throughout the manuscript: mg/mL, μl/mL, μmol/g: should be the same for all the concentrations used to be able to compare the data and connect between the amount found in leaves and the amount used in the assays and in between the assays. 

R. According to the reviewer’s comment, concentrations of phenylpropanoids and amino acids were changed and reported in mg g-1 (w/w) in Tables 1 and 2. Concentrations of crude extracts were modified, and now they are expressed in mg mL-1 (w/v). See along the text.

RR: Figure 1 is still µl/ml

Another serious concern is with table 5, where the OAI values listed seem to be completely wrong based on the formula provided in the methods, and the no. of laid eggs shows the opposite of what the OAI value implies and what the authors interpret in the text. 

R: We considered Oviposition Activity Index values ≥ 0.3 as significantly attractive and values ≤ -0.3 as significantly repellent according to Kramer and Mulla (1979). Thus, the calculations are correct. 

RR: This doesn’t answer my concern. Maybe I didn’t phrase it clear enough. I understand the equation OAI = (NT – NC) / (NT + NC)

And I also fully understand that positive OAI values indicate that more eggs were laid in the treated pots than in the control ones, so the tested extract was attractive. Conversely, negative OAI values indicate that more eggs were laid in the control pots than in the treated ones, so the tested extract was repellent. If it is around zero no statement can be made, therefore only values above 0.3 or below – 0.3 are evaluated.

My issue with the data presented in Table 5is: how were the OAI values in table 5 calculated with the numbers given in the column “No. of eggs” and the equation for OAI?

For all days of leaf extract testing the number of eggs laid with the extract are higher than the number of eggs laid with the control. How can the calculated OAI value then be negative? mathematically it can only be positive when using the given formula. Same for flower extract day 1.

Is it possible that the egg numbers for extract and control are given in the wrong order in Table 5?

R. The text was rephrased as follows: “Preparation of leaf- and flower- extracts for chromatographic analyses” line 115. 

RR: in the revised version I got it is still the same title.

L181. it is not comprehensibly what the concentrations of the extracts or compounds were that were mixed with water, which volume of extract/compound solution was mixed with which volume of water, and what the final concentrations in the assay were. 

R. The final volume was always 250 mL (see the WHO protocol mentioned in the following answer), and the concentrations tested are reported in lines 189- 194

RR: it is still not clear what the final solvent composition was. See my extended comment above

L186. For how long? and define what aged tap water means 

The treatments lasted 24 hours. Tap water was aged for 24 hours to let dissipate chlorine from it. 

RR: Please add all this information to the methods, not only in the answer to me.

L187. why this ratio? 

R. 50 μL mL-1 of methanol was its concentration in the 250 mL of vegetal extracts (see line 220). 

RR: still not clear, as I have doubts if all the extracts and single compound solutions were really all in 100% methanol. See all my comments above and in the manuscript.

L196. fourth stage 

R. Actually changed in “fourth-instar” for better coherence. 

RR: It is not changed in the revised version that is available to me

L323. what is that? not clear! where are these data shown, especially the ones for dependency on extract concentration. 

R. These are the output of the ANCOVA tests. Please, see Materials and Methods (lines 294-298). 

RR: please see my comment in the document 

L334. what is that? 

R. SE, Standard error. Please see the legend of the table. 

RR: Sorry, I might not have been clear. I meant “Intercept”. it should be explained in the legend. Please see my comment above

L334. what is this p-value? which significance is analysed here? 

R. The p-values is the one of the coefficients or the probability that, within a given model, the null hypothesis that a particular predictor’s regression coefficient is zero given that the rest of the predictors are in the model. They are based on the Wald test statistics of the predictors. 7

RR: please see the above comment

L365. if the data was produced then it should be shown here. which concentrations were tested? not specified anywhere; L367. was is scarce or not measurable as stated in the previous sentence? 

R. The concentrations tested are reported in Materials and Methods, lines 190-194. Mortality was only 5% at the highest concentration tested of amino acids and null in the lower ones. 

RR: these numbers should be specified on the text instead of “data no shown” Please add

L367. why was this enzyme chosen? 

One of the modes of action exerted by some botanical compounds, especially phenols, on insects is the inhibition of acetylcholinesterase, as supported by several pieces of evidence (references 42, 43, 52, 58). 

RR: this information should be added to the manuscript text

L399. why was this not tested against Culex? 

R. The reproductive ethology of C. pipiens is completely different from that of A. albopictus. The former lays its eggs in large natural bodies of water where it was possible to collect the eggs but not to test the extracts. The latter, on the other hand, lays its eggs in small containers in a peri-urban environment, like the ovitraps we used. 

RR: would be interesting to have this reasoning why not done for C. pipiens in the manuscript text

L400. first: where is the statistics for this? 

R. We considered Oviposition Activity Index values ≥ 0.3 as significantly attractive and values ≤ -0.3 as significantly repellent according to Kramer and Mulla (1979). 

2nd: there is a consistently much higher oviposition in the extract than in the control. how do you come to the conclusion that the extract is deterrent? it obviously is attractive? 

R. The reviewer is right. The extract is attractive the first two days after the exposure but repellent the others.

RR: This is not what the egg numbers for extract and control show in the table available to me. For leaf the egg numbers for extract are much higher than for control on day 3, 4 and 7! Did you possibly accidentally switch extract and control egg numbers in Table 5?

L409. why was this concentration chosen? what about other concentrations? 

R. All entomological tests were carried out on the basis of preliminary tests. For this in particular the concentration was chosen to be slightly below the LC50 for both extracts and for both mosquito species. 

RR: this should be stated as an explanation in the manuscript

L442. instead of the following lengthy and unclear listing of LC50 values for other compounds from the literature the authors should show these data was a comprehensive table clearly displaying the own and published LC50 values, including the plasnt, compound, and treated developmental stage, and in the text only summarize the results in comparison to their own 

R. As we are in the discussion we only reported a fraction of all the data available regarding the toxicity of vegetal extracts towards mosquito species, highlighting how some extract work better than ours, some in a comparable way, and some are less effective. For a really comprehensive table we should transform the section into a review, and this is not our purpose. 

RR: I’m not asking to include all published data in the table, just to display the data cited in the discussion in a table, because this section with the many numbers is tedious to read and it’s difficult to get the important points out. The manuscript would profit from listing all the numbers in a table, in comparison to the numbers of yyour study, and in the text only point out the relevant tendencies compared to your data. 

Comments on the Quality of English Language

I understood that the manuscript will be edited by an English language editing service. I hope that the parts where the meaning is still hard to get will then be clearer.

Author Response

Here are some of my strong concerns about the manuscript:

Throughout the experiments it is not clear which concentrations of the extracts or compounds were tested, in how many replicates, and what presented the data means (averages, means, SD, SEM, p-values etc).

  1. We agree with the reviewer that, being a multidisciplinary manuscript, the materials and methods section is quite long and complicated. The concentrations tested in the toxicity trials are reported in lines 190-194, and the repetitions were five. Each oviposition deterrence experimental unit consisted in a control and two different treatments (leaves and flowers extracts). The same control was utilized for the calculation of the OAI of both the treatments. So, the two treatments and the (one) control were replicated five times (see lines 265-269). For data meaning, please see the legend of the tables.

RR: I thank the authors for the additions made to the method section, that solved some of the questions I had. However, I’m still struggling to follow the experiments. Even after reading these sections multiple times and cross-referencing between the methods chapters I can’t figure out what was done in a way that I could repeat the experiments. Therefore, please add the missing information and / or make the descriptions clear.

Specifically, please address the following:

R1: Section: “Preparation of leaf- and flower-methanol extracts for chromatographic analyses”:

If I understand the text correctly then phenylpropanoid extracts were made in 100% methanol and amino acid extracts were made in 100% water (like also described in lines 96-99). However, the title speaks only about methanol extracts. Either the title, is not correct, or I don’t understand the text correctly. In either case, please edit to make it correct and/or make it clearly understandable what was done.

A1: The title was changed as follows: “Leaf and flower extracts from the dwarf elder (Sambucus ebulus): toxicity and repellence against cosmopolitan mosquito-borne diseases vectors”. Figure 1 and the subtitle were also modified accordingly.

R2: These extracts would then be analysed by HPLC as described in the following chapter, But are they also the “crude extracts” you refer to further below? Then it would be helpful to mention that here.

A2: We changed as suggested the text of M&M regarding the preparation of the extracts as “Preparation of leaf and flower extracts

A 100 mg fresh weight (FW) portion of S. ebulus leaves or flowers was homogenized in a mortar with 1 mL of 100% HPLC-grade methanol for phenylpropanoid extraction, while a 50 mg FW portion of leaf or flower tissues was mixed in HPLC water for amino acid extraction. Samples were then incubated overnight in the dark at 4 °C [35]. Extracts were centrifuged for 20 min at 16000 x g at 4 °C, and the respective supernatants were filtered through 0.2 μm Ministart SRT 15 filters (Sigma-Aldrich, Milan, Italy) and preserved in test tubes at -20 °C until chromatographic analyses. Both crude extracts were also tested in entomological bioassays against mosquitoes.

R3: Section “Determination and identification of phenylpropanoids and amino acids”

From the methods section above I concluded that the amino acid extracts were in water. But the amino acids standards used for chromatography were in methanol (line 177)? And methanolic solution means 100% methanol? Please specify in the methods

A3: Solvents were specified throughout the text (L123-126). Moreover, text has been modified according to the Reviewer’s comment (L164-169): “Alanine, arginine, aspartic acid, glycine, glutamic acid, glutamine, histidine, isoleucine, leucine, phenylalanine, serine, threonine, and valine were determined and quantified in each extract using the software above reported. Measurements were performed on three different extracts belonging to three biological replicates; two instrumental replicates were carried out for each extract”.

R4: I assume that the HPLC purified compounds were then used for the tests. Specify the solvents and concentrations in which they were obtained from the HPLC for use in the downstream tests.

A4: Solvents were specified throughout the text (L123-126).

R5. Section “Preparation of crude extracts and … compounds”:

The information provided here doesn’t allow to understand how these dilutions were done. First of all: “crude extracts” refers to the extracts from chapter “Preparation of leaf- and flower-methanol extracts for chromatographic analyses”? If yes, as stated above, please add this information to that chapter to make the cross-referencing easier. If the “crude extracts” are not what is described in that chapter, then the information on how the “crude extracts” were done, is completely missing or not deductible from this chapter. In this case please add in this chapter then how the crude extracts were made. Second: “methanol solutions of crude extracts were prepared”: how? Provide the details on what was mixed with what in which volumes. The initial crude extract was in what? 100% methanol or 100 % water? Specify! The Phenolic compounds and amino acids used here I assume come from the HPLC analysis? Please specifiy where they come from or how they were obtained. what was the concentration of these pure compound solutions and what was the solvents

The dilutions of the phenolic compounds were done in methanol (line 192/193). What were the amino acids solutions diluted in?

A5: The text has been changed according to the Reviewer’s comment as follows:

Preparation of leaf and flower extracts

A 100 mg fresh weight (FW) portion of S. ebulus leaves or flowers was homogenized in a mortar with 1 mL of 100% HPLC-grade methanol for phenylpropanoid extraction, while a 50 mg FW portion of leaf or flower tissues was mixed in HPLC water for amino acid extraction. Samples were then incubated overnight in the dark at 4 °C [35]. Extracts were centrifuged for 20 min at 16000 x g at 4 °C, and the respective supernatants were filtered through 0.2 μm Ministart SRT 15 filters (Sigma-Aldrich, Milan, Italy) and preserved in test tubes at -20 °C until chromatographic analyses. Both crude extracts will be also tested in entomological bioassays against mosquitoes.

R6: Section: “Toxicity tests on Aedes albopictus and Culex pipiens larvae”

“As a negative control, ten larvae were put in 250 mL of a solution of water and methanol at the concentration of 5 mg mL-1 (the same methanol concentration of the vegetal extracts).”

My question here is: this control makes only sense if all the dilutions of the leaf and flower extracts and of the single compounds were in 100% methanol, which, as all my questions to the above methods hopefully show, is not clear at all from the descriptions.

If they dilutions were not all in 100% methanol, then a corresponding control is missing!

A6: L 193-198. Text has been modified as follows: “Crude extracts at increasing concentrations ranging from 2.0 to 4.0 mg mL-1 were diluted in aged tap water. Likewise, phenolic compounds at increasing concentrations ranging from 1.0 to 5.0 mg mL-1 and amino acids ranging from 0.50 to 2.5 mg mL-1 were diluted in aged tap water. All the toxicity tests were preceded by preliminary assessments to select the most suitable concentrations to obtain mortality values > 0% and < 99%.

Regarding the controls (L 202-204) the text has been modified as follows “(same quantity of methanol in the extracts and phenylpropanoid solutions at the highest dose tested). For amino acids, controls were performed in 250 mL of aged tap water.”

I hope, from all these questions you can see how much I struggle to understand the details of what was done. Therefore, please improve this and provide all the requested information in a clearly describing way to meet the scientific standard for reproducibility of experiments.

The results descriptions and interpretations in the results text are often hard to comprehend, partly incomprehensible and in parts do not match the numbers presented in the tables.

  1. We thank the reviewer for the comment, but more details would be helpful in answering the question. However, if the reviewer intends the statistic parameters, please see the replies at lines 284-311.

RR: this was a general remark on the quality of the results interpretations and descriptions. The specific remarks you addressed below.

For tables 3 and 4 it is not clear what the column intercept and P refer to. the numbers don't match the numbers listed in the text. therefore it isn't clear what was analysed here and what these numbers mean, neither the numbers and p-values in the text, nor the numbers in the tables.

R7: The p-values is one of the coefficients or the probability that, within a given model, the null hypothesis that a particular predictor’s regression coefficient is zero given that the rest of the predictors are in the model. They are based on the Wald test statistics of the predictors.

RR: This isn’t clear to everyone. Please specify in the table caption what “Intercept” and “P” is, the same as you have done for LC50, CI and SE. All table columns should be explained if the column titles are not self-explanatory. Which they aren’t if the reader is not a statistics expert.

A7: Done, we indicated in the caption as requested the meaning of P and Intercept.

R8: Table and figure captions lack lots of information. All table and figure captions should include following information: on how many biological and technical replicates is the data based?, and how many individuals per replicate were used (if applicable)? What is presented in the tables and figures? i.e. averages, means, medians, SD or SEM?

  1. All the tables have a caption but also a legend under reported, where the reviewer can find the detail of what is presented in the table. For replicates, please see lines 264-267.

R: this doesn’t answer my point. It is not clear in all tables if the values given are means or medians, for table 1 and 2 it is also not specified if the indicated variances are the standard error or standard deviation. To better be able to judge the presented data it is also helpful to have the information on number of replicates and number or individuals per replicate (it makes it so much easier for the reader to not have to search this latter information in the methods)

Therefore, I repeat my unaddressed remark: every table of figure caption should contain information on what is presented, i.e. means, medians, SE or SD, and please also add information on no. of replicates and no. of individuals per replicate.

A8: We added the requested information in each table. The following info has been included in the captions of table 1 and 2: “Data are shown as means ± standard errors (n = 3 biological replicates for each tissue extract).” Additional info is also reported in L166-168. However, in case of table 3 and 4, the number of replicates and the number of individuals per replicate is not applicable as data are calculated by Probit regression.

R9. the methods lack too many details to be able to reproduce the experiments.

Inconsistent use of concentrations throughout the manuscript: mg/mL, μl/mL, μmol/g: should be the same for all the concentrations used to be able to compare the data and connect between the amount found in leaves and the amount used in the assays and in between the assays.

  1. According to the reviewer’s comment, concentrations of phenylpropanoids and amino acids were changed and reported in mg g-1 (w/w) in Tables 1 and 2. Concentrations of crude extracts were modified, and now they are expressed in mg mL-1 (w/v). See along the text.

RR: Figure 1 is still µl/ml

A9: We changed the units in Fig. 1 (now figure 2). Thanks for the suggestion.

R10. Another serious concern is with table 5, where the OAI values listed seem to be completely wrong based on the formula provided in the methods, and the no. of laid eggs shows the opposite of what the OAI value implies and what the authors interpret in the text.

R: We considered Oviposition Activity Index values ≥ 0.3 as significantly attractive and values ≤ -0.3 as significantly repellent according to Kramer and Mulla (1979). Thus, the calculations are correct.

RR: This doesn’t answer my concern. Maybe I didn’t phrase it clear enough. I understand the equation OAI = (NT – NC) / (NT + NC)

And I also fully understand that positive OAI values indicate that more eggs were laid in the treated pots than in the control ones, so the tested extract was attractive. Conversely, negative OAI values indicate that more eggs were laid in the control pots than in the treated ones, so the tested extract was repellent. If it is around zero no statement can be made, therefore only values above 0.3 or below – 0.3 are evaluated.

My issue with the data presented in Table 5is: how were the OAI values in table 5 calculated with the numbers given in the column “No. of eggs” and the equation for OAI?

For all days of leaf extract testing the number of eggs laid with the extract are higher than the number of eggs laid with the control. How can the calculated OAI value then be negative? mathematically it can only be positive when using the given formula. Same for flower extract day 1.

Is it possible that the egg numbers for extract and control are given in the wrong order in Table 5?

A10: We sincerely apologise to the reviewer for this oversight. In the previous revision, we checked the final results but not the position of the lines. The reviewer is absolutely right, the lines were reversed, and we put them now in a right way.

R11. The text was rephrased as follows: “Preparation of leaf- and flower- extracts for chromatographic analyses” line 115.

RR: in the revised version I got it is still the same title.

A11: Once again, we apologise for the oversight

R12: L181. it is not comprehensibly what the concentrations of the extracts or compounds were that were mixed with water, which volume of extract/compound solution was mixed with which volume of water, and what the final concentrations in the assay were.

  1. The final volume was always 250 mL (see the WHO protocol mentioned in the following answer), and the concentrations tested are reported in lines 189- 194.

RR: it is still not clear what the final solvent composition was. See my extended comment above

A12. The doses are now reported at L193-198.

R13: L186. For how long? and define what aged tap water means

The treatments lasted 24 hours. Tap water was aged for 24 hours to let dissipate chlorine from it.

RR: Please add all this information to the methods, not only in the answer to me.

A13: We added the information in M&M as suggested (see L206)

R14: L187. why this ratio?

  1. 50 μL mL-1 of methanol was its concentration in the 250 mL of vegetal extracts (see line 220).

RR: still not clear, as I have doubts if all the extracts and single compound solutions were really all in 100% methanol. See all my comments above and in the manuscript.

A14: Text was changed according to the previous Reviewer’s comments, please see L 202-204 “(same quantity of methanol in the extracts and phenylpropanoid solutions at the highest dose tested). For amino acids, controls were performed in 250 mL of aged tap water.”

R15: L196. fourth stage

  1. Actually changed in “fourth-instar” for better coherence.

RR: It is not changed in the revised version that is available to me

A15. We apologize for the oversight. We have now corrected it.

R16: L323. what is that? not clear! where are these data shown, especially the ones for dependency on extract concentration.

  1. These are the output of the ANCOVA tests. Please, see Materials and Methods (lines 294-298).

RR: please see my comment in the document

A16: We specified in the text (L .347) the statistical analysis done (ANCOVA)

R17: L334. what is that?

  1. SE, Standard error. Please see the legend of the table.

RR: Sorry, I might not have been clear. I meant “Intercept”. it should be explained in the legend. Please see my comment above

A17: Done, we added the information in the captions.

R18: L334. what is this p-value? which significance is analysed here?

  1. The p-values is the one of the coefficients or the probability that, within a given model, the null hypothesis that a particular predictor’s regression coefficient is zero given that the rest of the predictors are in the model. They are based on the Wald test statistics of the predictors. 7

RR: please see the above comment

A18: Done, we added the information in the captions.

R19: L365. if the data was produced then it should be shown here. which concentrations were tested? not specified anywhere; L367. was is scarce or not measurable as stated in the previous sentence?

  1. The concentrations tested are reported in Materials and Methods, lines 190-194. Mortality was only 5% at the highest concentration tested of amino acids and null in the lower ones.

RR: these numbers should be specified on the text instead of “data no shown” Please add

A19. We added amino acids mortality data as suggested (L 395-395)

R20: L367. why was this enzyme chosen?

One of the modes of action exerted by some botanical compounds, especially phenols, on insects is the inhibition of acetylcholinesterase, as supported by several pieces of evidence (references 42, 43, 52, 58).

RR: this information should be added to the manuscript text

A20. We added the information as suggested (L 211-213)

R21: L399. why was this not tested against Culex?

  1. The reproductive ethology of C. pipiens is completely different from that of A. albopictus. The former lays its eggs in large natural bodies of water where it was possible to collect the eggs but not to test the extracts. The latter, on the other hand, lays its eggs in small containers in a peri-urban environment, like the ovitraps we used.

RR: would be interesting to have this reasoning why not done for C. pipiens in the manuscript text

A21: We added the explication as suggested, see L 245-249

R22: L400. first: where is the statistics for this?

  1. We considered Oviposition Activity Index values ≥ 0.3 as significantly attractive and values ≤ -0.3 as significantly repellent according to Kramer and Mulla (1979). 8

2nd: there is a consistently much higher oviposition in the extract than in the control. how do you come to the conclusion that the extract is deterrent? it obviously is attractive?

  1. The reviewer is right. The extract is attractive the first two days after the exposure but repellent the others.

RR: This is not what the egg numbers for extract and control show in the table available to me. For leaf the egg numbers for extract are much higher than for control on day 3, 4 and 7! Did you possibly accidentally switch extract and control egg numbers in Table 5?

A22. The reviewer is absolutely right, the lines were reversed, and we put them now in a right way. We corrected the table. Thanks for the suggestion.

R23: L409. why was this concentration chosen? what about other concentrations?

  1. All entomological tests were carried out on the basis of preliminary tests. For this in particular the concentration was chosen to be slightly below the LC50 for both extracts and for both mosquito species.

RR: this should be stated as an explanation in the manuscript

A23. Done, please see L 253-254

R24: L442. instead of the following lengthy and unclear listing of LC50 values for other compounds from the literature the authors should show these data was a comprehensive table clearly displaying the own and published LC50 values, including the plasnt, compound, and treated developmental stage, and in the text only summarize the results in comparison to their own

  1. As we are in the discussion we only reported a fraction of all the data available regarding the toxicity of vegetal extracts towards mosquito species, highlighting how some extract work better than ours, some in a comparable way, and some are less effective. For a really comprehensive table we should transform the section into a review, and this is not our purpose.

RR: I’m not asking to include all published data in the table, just to display the data cited in the discussion in a table, because this section with the many numbers is tedious to read and it’s difficult to get the important points out. The manuscript would profit from listing all the numbers in a table, in comparison to the numbers of yyour study, and in the text only point out the relevant tendencies compared to your data.

A24: Done, we added a table as suggested (see Table 6, L461). We hope that the discussion is now easier to read.

R25: Comments on the Quality of English Language

I understood that the manuscript will be edited by an English language editing service. I hope that the parts where the meaning is still hard to get will then be clearer.

A25: Thank you for your understanding. Revision of English is done by the editorial system, and if the English is not deemed good enough by the editor, we will use the paid service.

We would like to thank the reviewer for his patience and for his authoritative comments, which have certainly improved the quality of the manuscript.

Round 3

Reviewer 2 Report

Comments and Suggestions for Authors

Dear authors,

thank you again for addressing all my suggestions and concerns. A lot of things are now much clearer, which increases the significance and value of your study for the research community. I'm happy to agree to the publication of the study in the current version. 

Comments on the Quality of English Language

The authors confirmed already that the English language will be further improved

Author Response

In table 5, please double check the results of leaf extract. For day 3, a significant repellent effect is identified (-0.67) while the number of eggs in the extract is higher (17 +/- 9) than in the control (3.3 +/- 3.3). Please check the results for day 1 & 2 as well.

Done, we verify and correct the results of the days 1, 2 and 3 (evidenced in pink)

Lines 434-444: As I already noted, in the first paragraph of the discussion the findings of the current study regarding chemical analysis should be mentioned with respect to other studies. The text of this paragraph could be limited as well.

The first paragraph of discussion has been modified according to the Reviewer’s suggestion. (L441-453).

Lines 454-455: The sentence could be reworded as follows “…, we present in Table 6 the results obtained from our screening by comparing our LC50 values to those reported by other authors”.

We reworded the sentence as indicated (evidenced in pink).

The text in lines 504-512 is more suitable for the “conclusions” section.

We moved the text indicated in the conclusion section as suggested (evidenced in pink).